# Recrystallization of Hot-Rolled 2A14 Alloy during Semisolid Temperature Annealing Process

**DOI:** 10.3390/ma16072796

**Published:** 2023-03-31

**Authors:** Yingze Liu, Jufu Jiang, Ying Zhang, Minjie Huang, Jian Dong, Ying Wang

**Affiliations:** 1School of Materials Science and Engineering, Harbin Institute of Technology, Harbin 150001, China; 2School of Mechatronics Engineering, Harbin Institute of Technology, Harbin 150001, China

**Keywords:** semisolid annealing, 2A14 aluminum alloy, hot-rolled, EBSD, TEM, recrystallization

## Abstract

In this study, in order to provide proper parameters for the preparation of semisolid billets, the semisolid annealing of hot-rolled 2A14 Al alloy was investigated. The microstructure was characterized by X-ray diffraction (XRD), scanning electron microscopy (SEM) with an X-ray energy dispersive spectrometer (EDS) and electron backscattered diffraction (EBSD), and scanning transmission electron microscopy (STEM). The XRD results showed that, with an increase in temperature, the θ-Al2Cu equilibrium gradually dissolved in the matrix. The EDS results of SEM and STEM showed a coarse θ-Al2Cu phase, ultrafine precipitate Al(MnFeSi) or (Mn, Fe)Al6 phase, and atomic clusters in the microstructure. The EBSD results showed that the recrystallization mechanism was dominated by continuous static recrystallization (CSRX), homogeneous nucleation occurred when the sample was heated to near solidus temperature, and CSRX occurred at a semisolid temperature. In the process of recrystallization, the microtexture changed from the preferred orientation to a random orientation. Various experimental results showed that static recrystallization (SRX) occurred at a semisolid temperature due to the blocking effect of atomic clusters on the dislocation slip, and the Zener drag effect of fine precipitates on low-angle grain boundaries (LAGBs) disappeared with melting at a semisolid temperature.

## 1. Introduction

Aluminum alloys are still used in the field of aerospace. They have excellent comprehensive mechanical properties and unparalleled economic applicability, even in the face of competition from magnesium alloys, titanium alloys, and composites [1,2,3]. In addition to improving the alloy composition, the innovation of thermomechanical processing (TMP) is important. Having reasonable TMP can not only save materials and reduce costs, it can also help to ensure the reasonable distribution of the polycrystalline texture and achieve the purpose of strengthening the mechanical properties. Semisolid metal processing (SSM) is a mature metal near net shape forming technology [4]. It is widely used in the formation of complex-shaped aluminum alloy parts. The key step in SSM is to prepare semisolid billets. The strain-induced melt activation (SIMA) method and the recrystallization and partial melting (RAP) method are two traditional methods used to fabricate semisolid billets [5]. The method of preparing semisolid billets can be summarized as pre-deformation (PD) of the material, and then semisolid isothermal treatment (SSIT) is used to prepare the semisolid billet. According to different PD methods, several semisolid billet preparation methods can be derived [5]. Binesh and Aghaie-Khafri [6] introduced the repetitive upsetting extrusion (RUE) method for the PD of modified SIMA, increasing the microstructure of semisolid 7075 alloy. Fu et al. [7] investigated the coarsening behaviors of 7075 aluminum alloy during the equal channel angular pressing (ECAP)-based SIMA method. The introduction of severe plastic deformation (SPD) as the predeformation method for preparing semisolid billet is a useful way to refine the grain of semisolid billet, but it also complicates the process and increases the cost. In this paper, to simplify the semisolid billet preparation processing, a commercial hot-rolled 2A14 thick plate was used for the original material. Instead of additional deformation processing, it was directly subjected to SSIT. This method is known as the wrought aluminum alloy directly semisolid thermal treatment (WADSSIT) method [8].

Due to their excellent mechanical performance and structural carrying capacity and good weldability, 2A14 aluminum alloys are used as the material for key structural parts of aircrafts, such as wheels and holders with complex shapes [9]. Nassiraei et al. [10] proposed a novel method to determine the ultimate capacity of collar plate reinforced weld X-joints at elevated temperatures. Lan et al. [9] found that cold deformation prior to artificial aging will promote the precipitation of phases θ″, θ′, and θ (Al_2_Cu) in 2A14 alloy, thereby shortening the peak aging time. Wang et al. [11] investigated multidirectional forging (MDF) on the 2A14 aluminum alloy. The transmission electron microscopy (TEM) results revealed that second-phase particles dynamically precipitated during deformation, inhibiting the dislocation motion and thus increasing the yield strength of the alloy. Wang et al. [12] found that higher MDF temperatures increase the solubility and diffusion efficiency of the solute atoms, resulting in fine and spherical second-phase particles, which improves the mechanical properties. Research on the 2A14 alloy focused on forging [9,11,12], but there have been no reports on SSM in the 2A14 alloy. For parts with complex shapes, SSM has the advantages of saving raw materials and reducing the cost of subsequent machining. In addition, research on SSM has focused more on 7xxx series aluminum than 2xxx series aluminum alloy [7,13,14,15]. However, different series of aluminum alloys have significant differences in their phase compositions, semisolid ranges, and recrystallization mechanisms. Therefore, it is significant to investigate the SSM of the 2A14 alloy.

The recrystallization phenomenon occurs in various TMPs [16]. It has an important influence on the microstructure, texture evolution, and properties of various TMPs [17]. Recrystallization also plays an important role in the preparation of semisolid billets. In the predeformation stage, dynamic recrystallization (DRX) may take place so that the deformation energy is stored in the material in the form of high- (HAGBs) and low-angle grain boundaries (LAGBs). In the temperature rising stage of SSIT, static recrystallization (SRX) may occur, where the deformed grains turn into uniform equiaxed grains. Generally, discontinuous static recrystallization (DSRX) may occur during SSIT with clear nondeformation nuclei formation and growth [6,15,18]. SRX can also take place homogeneously without a clear nucleation and growth stage, which is labelled continuous static recrystallization (CSRX). CSRX is a common phenomenon in aluminum alloys with a particle-stabilized subgrain structure [19]. However, there have been no reports on CSRX in the process of semisolid preparation. On one hand, because the research focus of semisolid billet preparation is often on grain refinement, roundness, and microstructure coarsening behavior, recrystallization behavior in the temperature rising process is often ignored [15,20]. On the other hand, increasing the degree of predeformation to refine the microstructure is also a more common research method, which may affect the occurrence of CSRX. The SRX behavior will affect the coarsening behavior and texture, providing technical guidance for the preparation of semisolid billets. Therefore, it is necessary to study the recrystallization behavior in the temperature rising stage.

Electron backscatter diffraction (EBSD) combined with software, such as channel 5 or Mtex, is suitable for the characterization of polycrystalline materials. The kernel average misorientation (KAM), grain reference orientation deviation (GROD), and grain orientation spread (GOS) are powerful tools that can be used to characterize the recrystallization behavior. The orientation distribution function (ODF) can be used to characterize the microtexture of the corresponding scanning area. However, EBSD is limited by the resolution range of its coupled scanning electron microscope and cannot characterize the interactions between dislocated and precipitated particles. Transmission electron microscopy (TEM) can be used to tackle this problem. In this paper, the SRX behavior and microtexture of the 2A14 Al alloy in the temperature rising stage during WADSSIT were investigated systemically through EBSD and TEM technology. The reasons for the effects of CSRX on spheroidization and coarsening in the SSIT process are discussed.

## 2. Materials and Methods

### 2.1. Starting Material

The wrought 2A14 aluminum alloys used in this study were purchased from the Northeast Light Alloy Co., Ltd. of China. The supply status was a hot rolled medium-thick plate, 50 mm in size, with a cumulative deformation of 90% at 400 °C. The chemical composition was tested by X-ray fluorescence (XRF), as shown in Table 1. As shown in Figure 1, the semisolid temperature ranged from 512 °C to 663 °C as tested via differential scanning calorimetry (DSC, STA449F3). The scanning speed was 10 °C/min. The solid fraction $f_{s}$ was determined by Equation (1)
(1)$$f_{s}\left(T\right)=1-\left[\frac{\left(H-H_{solidus}\right)-C_{p}\times (T-T_{solidus})}{\left(H_{liquidus}-H_{solidus}\right)-C_{p}\times (T_{liquidus}-T_{solidus})}\right],$$
where *H* is the enthalpy, *T* is the temperature, *C_p_* is the heat capacity of the alloy, and *L* is the latent heat of fusion. The solid fraction vs. temperature curve is shown in Figure 1.

### 2.2. Recrystallization Process

The WADSSIT method was used to fabricate semisolid billets of the 2A14 alloy. It includes a temperature rising stage and a temperature holding stage, as shown in Figure 2. The recrystallization process occurred during the temperature rising stage. Cuboid samples, 8 × 10 × 12 mm^3^ in size, were cut from the hot-rolled plate, and the rolling direction (RD), normal direction (ND), and transverse direction (TD) were distinguished by the side length of the sample. The samples were continuously heated from room temperature to the given temperatures in a muffle furnace. When the temperature reached 480 °C, 505 °C, 530 °C, 555 °C, 580 °C, 600 °C, and 620 °C, the samples were quenched in cold water immediately to freeze the microstructure. In the temperature holding stage, the holding time was set to 5 min, 10 min, 15 min, 20 min, 25 min, and 30 min.

### 2.3. Characterization of Microstructure

For the optical microscopy (OM) observation, the equipment used was an Olympus GX71. The samples were ground with 100, 400, 800, 1000, and 2000 grit papers and then polished on a gold velvet disc sprayed with 1 μm diamond powder at 1200 rpm for 5 min, before being chemically etched with Keller’s reagent. The chemical composition of the Keller’s reagent was an aqueous solution with 2.5%HNO_3_, 1.5%HCl, and 1%HF, and the etching time was 25 s.

For EBSD characterization, the equipment used was a Zeiss Supra 55 scanning electron microscope (SEM) equipped with the EBSD function and X-ray energy dispersive spectrometer (EDS). The samples underwent the same ground and mechanical polishing process as the OM samples. Then, in order to remove the influence of the pseudo stress layer introduced by mechanical polishing on the EBSD test, the samples were ion polished using Leica RES101 with a voltage of 6.5 V and a polishing time of 1 h. In order to obtain as accurate EBSD information as possible, the scanning step was set to 2 microns, and the scanning range of each sample was set to 470 × 350 = 164,500 scanning points. Channel 5 version 5.0.9.0 and Mtex version 5.4.0 analysis software was used to analyze the EBSD data.

For the transmission electron microscope (TEM) characterization, the equipment used was the FEI Talos F200X. The samples were cut to 300 μm thickness by wire cutting. Then, the samples were polished to a thickness of 30 μm with 2000 grit SiC sandpaper and a screw micrometer. Then, 3 mm diameter discs were punched out from the samples with a punch. The discs were etched by argon ion with a precision ion polishing system (Gatan 695) to a thickness of about 100 nm.

## 3. Results

### 3.1. Microstructure of the Raw Material

Figure 3a–c shows the OM images of the hot-rolled plate on the ND-RD plane, TD-ND plane, and TD-RD plane, respectively. A schematic diagram of the hot-rolled plate direction is shown in Figure 3d. As the accumulated deformation in the hot rolling process reached 90%, DRX occurred under hot large plastic deformation, resulting in a large number of dislocation entanglements and LAGBs. There were a lot of defects in the crystal, which made the energy in the crystal close to that at the grain boundary. Therefore, it was difficult to distinguish the grain boundary contrast etched by the Keller’s reagent under the optical microscope. However, due to the different grain orientation, the corrosion resistance was different, and the contrast of some grains could be distinguished. In Figure 3a, the grains are elongated along the RD. In Figure 3b, the grain contrast is almost invisible. In Figure 3c, in addition to elongated grains along the RD, there are also fine equiaxed grains with a size of about 50 microns. Due to the large cumulative deformation and limited by the analytical ability of the SEM-EBSD equipment used, the resolution of EBSD resulted in the raw material on the ND-RD, TD-RD plane being unsatisfactory. In order to study the evolution of the microtexture and eliminate the analysis of the pseudo results due to insufficient resolution, an EBSD test of the raw material was carried out on the TD-ND plane, and the EBSD-IPF results are shown in Figure 3d. In MTEX version 5.4.0 software, the LAGB threshold for identifying grains was set to 15°, which was outlined by black lines. On the TD-ND plane, the microstructure was composed of huge, elongated grains and small, fine grains. In large grains, the color of the IPF differed at different positions, indicating a certain degree of misorientation at different positions in the grains.

### 3.2. EBSD Analysis Results during the Recrystallization Process

Figure 4a–c shows the EBSD-IPF images of the 2A14 Al alloy on the RD-TD plane during annealing to 505 °C, 555 °C, and 580 °C, respectively. The corresponding IPF grain orientation ruler is shown in Figure 4d. With an increase in the annealing temperature, the grain size increased gradually. When annealing to 505 °C and 555 °C, the color difference in a single grain was large, which indicates that there were still many defects in the grain and that LAGBs existed. When annealing to 580 °C, the color in a single grain was relatively consistent, indicating that recovery and recrystallization occurred during the process of increasing the temperature and that the defects in the grain were gradually reduced.

In order to further study the evolution of the grain boundaries during recrystallization annealing, a grain boundary diagram and grain boundary density diagram on the RD-TD plane were drawn, as shown in Figure 5. Due to the limited resolution accuracy of the EBSD test and the magnification of the SEM, the threshold of LAGB recognition was set to 2°, and subgrain boundaries of less than 2° were ignored. When the annealing temperature reached 505 °C (Figure 5a), the microstructure was densely covered with LAGBs, the peak relative density of LAGBs reached 14, and the relative density of LAGBs in most areas was higher than 12 (Figure 5d). With the increase in the annealing temperature to 555 °C (Figure 5b), the grain size became larger; in other words, there were fewer HAGBs. Some grains with low-density grain boundaries appeared in the structure, which indicates that the recovery phenomenon of the structure occurred alongside the recrystallization process. The relative density peak of the grain boundary was still 14 (see Figure 5e), while the relative density of the grain boundary remained at a high level in most areas, and only low-density grain boundary areas appeared in local areas. With an increase in the annealing temperature, the grain boundaries will migrate, and the migration of grain boundaries will reduce the density of the grain boundaries in the crystal. When annealing at 505 °C, the migration of LAGBs was obviously blocked. With an increase in the temperature to 555 °C, the grain boundary density decreased in some areas, indicating that this hindrance was disappearing. When the annealing temperature reached 580 °C (Figure 5c), the grains grew further, and the LAGBs in the grains were greatly reduced, leading to their disappearance. The grain boundary density figure (Figure 5f) shows that the peak relative density of the grain boundary decreased to 1.4 at 580 °C, which is one tenth of that at 505 °C and 555 °C. At the same time, the relative grain boundary density in most areas was less than 1.

In the process of annealing, in addition to the evolution of grain boundaries, the evolution of dislocations is an interesting phenomenon. Generally speaking, with an increase in the annealing temperature, dislocations will slip and climb, different dislocations will annihilate, and the same dislocations will be directionally arranged to form dislocation walls (subgrain boundaries). Due to the annihilation of different dislocations, the dislocation density will decrease, and this is often called the metal recovery phenomenon. Unfortunately, the resolution scale of EBSD cannot reach the dislocation level, but the change trend of the dislocation density can be qualitatively analyzed by the theoretical calculation.

The kernel average misorientation results of the 2A14 Al alloy on the RD-TD plane during annealing to 505 °C, 555 °C, and 580 °C are shown in Figure 6a–c, respectively. When the annealing temperature reached 505 °C (Figure 6a), the peak value of KAM reached 15, and the KAM value in most areas was greater than 10, indicating that the geometrically necessary dislocation (GND) density in the structure was high. With the increase in temperature to 555 °C (Figure 6b), the peak value of KAM was still 15, indicating that the GND density of the high-density dislocation regions had not decreased. The KAM value of most areas decreased to below 10, and in some areas, KAM values lower than 5 appeared. Compared with Figure 5b,e, these low-density dislocation regions coincided with the low-density grain boundary regions. When the annealing temperature reached 580 °C (Figure 6c), the peak value of KAM decreased to 10 and the KAM value of most areas decreased below 3, which shows that the GND density in most regions decreased, and only the GNDs density near the HAGBs was higher. The annihilation of reverse dislocations will reduce the dislocation density. In order to undergo annihilation, dislocations first need to slip and climb. When annealed to 555 °C, the dislocations in some regions could slip and annihilate, reducing the dislocation density and forming a low-density dislocation region. The decrease in the KAM value was not obvious in most areas, indicating that the slip of dislocation was hindered. It is speculated that the slip of dislocation was hindered due to the existence of a dispersed precipitation phase, but this needed to be further confirmed by the energy dispersive spectrum (EDS) test.

Apart from the evolution of dislocations, the recrystallization can also be characterized more intuitively by the EBSD results. For each individual grain separated by the HAGBs, the average orientation of the grain can be obtained by calculating the orientation of each pixel in the grain. The orientation of each pixel has a deviation from the average orientation of the grain. The grain orientation spread (GOS) value can be obtained by averaging the deviations of all pixels in the grain from the average orientation. Generally speaking, grains with a GOS value below 1.8° are regarded as recrystallized grain structures, grains with a GOS value between 1.8 and 3 are regarded as subgrain structures, and grains with a GOS value over 3 are regarded as deformed grain structures. The GOS figures of the 2A14 Al alloy on the RD-TD plane after annealing to 505 °C, 555 °C, and 580 °C are shown in Figure 6d–f, respectively. When annealed to 505 °C (Figure 6d), the peak value of GOS reached 38, the GOS value of most grain regions exceeded 10, and the GOS value of almost all grains was higher than 3, indicating that it was a deformed structure at this time. When annealed to 555 °C (Figure 6e), the peak value of GOS decreased to 23, the GOS value of most grain regions decreased to 8, and the GOS values of some grain regions were lower than 3, indicating that some substructures appeared. When annealed to 580 °C (Figure 6f), the peak value of the GOS decreased to 7, the GOS values of most grain regions were lower than 3, and the GOS values of half of the grain regions were lower than 1.8, indicating that recrystallization has basically been completed, and there were a few substructures and deformed structures.

In addition, there was a phenomenon of texture in the hot rolling structure. The microtexture of the corresponding region can be characterized by analyzing the EBSD results. Figure 7a–d shows the pole figure of the 2A14 alloy at the hot-rolled stage after annealing to 505 °C, 555 °C, and 580 °C, respectively. The pole figures show the distribution of the texture in {100}, {110}, and {111}. There was an obvious texture phenomenon in the hot rolling structure (Figure 7a), and the intensity of the texture was the highest at {100}, while the peak intensity of the texture reached 9.36. With an increase in the annealing temperature, the peak intensity of the texture decreased. The peak intensity of the texture was decreased to 5.56, 3.56, and 3.01 when annealed to 505 °C, 555 °C, and 580 °C, respectively. At the same time, with an increase in the annealing temperature, the preferred orientation gradually changed to a random orientation.

In order to calibrate the type of texture, the corresponding orientation distribution function (ODF) maps were drawn. The ODF maps display a constant value of φ_2_ at an interval of 5°. ODF maps of the hot-rolled state after annealing to 505 °C, 555 °C, and 580 °C are shown in Figure 8a–d, respectively. The intensity of the texture was the highest in the hot-rolled state, reaching 13.208. With an increase in the annealing temperature, the intensity of the texture gradually decreased. The intensity of the texture decreased to 5.539, 4.549, and 4.437 after annealing to 505 °C, 555 °C, and 580 °C, respectively. Table 2 shows some of the common texture types that appeared in this study [21,22]. According to the method of formation, the texture can be divided into a deformation texture and recrystallization texture [23,24,25]. A mixed texture appeared in the hot-rolled state (Figure 8a). The texture with the highest intensity was the recrystallized P texture with {011} <122>. The second-highest intensity texture was the recrystallized brass texture with {236} <385>. The three deformation textures in order of intensity were the copper texture with {112} <111>, the brass texture with {011} <211>, and the S texture with {123} <634>, respectively. With an increase in the annealing temperature, these textures appeared in the subsequent annealing process, but the intensity of the texture changed. The peak intensity of the ODF maps gradually decreased, indicating that the texture intensity decreased with an increase in the annealing temperature, whether the texture was a deformation texture or a recrystallization texture. In addition, with an increase in the annealing temperature, the texture type changed from the mixed texture type of the deformation texture and recrystallization texture to the mixed texture type dominated by the recrystallization texture. These results also show that, in the process of recrystallization, the microstructure gradually changed from the preferred orientation to a random orientation, but the recrystallization texture was inherited in the process of transformation, and the strength gradually decreased.

In addition, the analysis of the misorientation axis can also explain the change in texture during recrystallization annealing. Figure 9 shows the misorientation axis figure and the corresponding distribution of the misorientation axis figure after annealing to 505 °C, 555 °C, and 580 °C, respectively. With an increase in the temperature, the density of the misorientation axis gradually decreased. The highest intensity of the misorientation axis was [001], and this occurred after annealing to 505 °C. When the annealing temperature rose to 555 °C, the highest intensity of misorientation axis changed to [$\bar{1}$11]. The misorientation axis in [001] still existed, but the intensity decreased greatly. When the annealing temperature rose to 580 °C, the misorientation axis in [001] disappeared. In general, the density of the misorientation axis decreased, which is consistent with the previous results, indicating that the microstructure changed from the preferred orientation to a random orientation during recrystallization annealing, but the recrystallization texture was retained and the deformation texture tended to disappear.

### 3.3. Second-Phase Particles in the 2A14 Alloy

The EBSD results show that recovery and recrystallization occur during recrystallization annealing. At 555 °C, the LAGBs in some grains can migrate, while the LAGBs in most grains show no obvious migration. It is inferred that the migration of LAGBs is hindered by the second-phase particles, so it is necessary to detect the phase and element distribution in the 2A14 Al alloy. Figure 10 shows the X-ray diffraction (XRD) patterns of the 2A14 Al alloy in the hot-rolled state after annealing to 505, 555, and 580 °C, respectively. The diffraction peaks of the four patterns occurred at the 2θ phase of 38.47°, 44.74°, 65.13°, 78.23°, and 82.46°, which corresponds to the diffraction of the (111), (200), (220), (311), and (222) planes of the Al matrix phase, respectively. Weaker diffraction peaks of the four patterns occurred at the 2θ phase of 20.62°, 37.87°, 45.59°, 47.33°, and 47.81°, which corresponded to the diffraction of the (110), (211), (112), (310), and (202) planes of the θ-Al2Cu phase, respectively. It is difficult to observe phases other than the Al matrix and θ phase in the XRD patterns, but this does not mean that there were only two phases in the alloy. The absolute proportions of other phases were too low to be observed by XRD. The characterization of other phases is given in the SEM and TEM results. The diffraction peak intensity of the θ phase was the same at that in the hot-rolled state after annealing to 505 °C, but it decreased after annealing to 555 °C, and nearly disappeared after annealing to 580 °C, which indicates that the θ phase was dissolved in the Al matrix; that is, the solid solution phenomenon occurred at 555 °C. This is consistent with the DSC results, which showed a dissolution endothermic peak of the θ phase at 520 °C (Figure 1). In addition, the relative intensities of diffraction peaks on different crystal planes of the Al matrix also evolved during annealing. For example, the diffraction peak of the Al matrix at the 2θ phase of 38.47° corresponding the (111) plane of Al matrix first increased and then decreased with the increase of annealing temperature. The diffraction peak of Al matrix at the 2θ phase of 82.46°, corresponding to the (222) plane of the Al matrix almost disappearing after annealing to 580 °C. The results show that the microtexture evolved during the annealing process, and the crystal plane evolved from the preferred orientation to a random orientation, which is consistent with the results of the PF (Figure 7) and ODF (Figure 8) diagram in EBSD.

Figure 11 shows the distribution of elements in the 2A14 Al alloy at different scales. Figure 11a shows the high-angle annular dark-field (HAADF) scanning transmission at a higher magnification for the 2A14 Al alloy after annealing to 555 °C. In the figure, precipitates are dispersed in the subgrains. Figure 11b is the corresponding scanning map distribution diagram of the elements. It can be seen from Figure 11b that the second-phase particles are dispersed in the subgrains. Among them, the larger dispersed precipitate phase is the Fe-Mn dispersed phase, and the smaller dispersed phases are the Si phase, Mg-Zn phase, and G-P zone formed by very small-scale enrichment of the Cu element. The size of these dispersed precipitates was small enough to hinder the migration of subgrain boundaries, LAGBs, and dislocation slip during recrystallization annealing, which is usually called the pinning effect of second-phase particles. Figure 11c shows a backscattered electron (BSE) map of the SEM image at a lower magnification. The contrast of BSE can well reflect the differences in the atomic numbers of elements. Scanning at a low magnification showed that coarse second phases were distributed along the HAGBs. According to the corresponding element scanning distribution diagram, the coarse second phases were mainly the Cu rich phase, Fe-Mn phase, and Si rich phase.

### 3.4. Microstructure with a Smaller Hierarchy of Scale

The characterization of EBSD is limited to the relatively low magnification of SEM, and the smallest hierarchy of crystal defect that can be distinguished is the LAGBs. EBSD provides weak results for the characterization of the microstructure at the scale of the dislocation substructure. Therefore, in order to better understand the evolution of smaller defects in the annealing process, it is necessary to characterize smaller defects such as subgrains and subgrain boundaries (dislocation walls) by TEM. Although in the TEM at higher magnification, due to the small field of view, the characteristics of the local area cannot reflect the characteristics of the overall material, a more typical area can still be selected to represent the characteristics of the overall material. Figure 12a,b shows the TEM image of 2A14 Al annealed to 505 °C with different levels of magnification. It can be seen that there were a large number of slip bands, high-density dislocations, and dislocation walls (subgrain boundaries) formed by the directional arrangement of dislocations in the subgrains. The cells formed by grain subdivision or fragmentation were separated by dislocation walls. This shows that there was no obvious recrystallization phenomenon when annealing to 505 °C, and even the annealing recovery phenomenon was not obvious. In addition, insoluble particles in the cells and the phenomenon of pinning dislocations by insoluble particles were observed. A typical TEM image of 2A14 Al annealed to 555 °C is shown in Figure 12c. The dislocation density in most regions was lower than that after annealing to 505 °C, and some precipitated second phases were seen in the cell and near the subgrain boundary. A typical TEM image of 2A14 Al annealed to 580 °C is shown in Figure 12d. The sizes of the precipitated second phases were smaller than those for the materials annealed to 555 °C. At 580 °C, it is obvious that the second phases should have been dissolved into the matrix. Therefore, the precipitated phases shown in Figure 12d are finer secondary precipitated phases that are produced after sample preparation and quenching to room temperature. This was a phenomenon caused by sample preparation problems. The defects of dislocations on subgrain boundaries were minimized after annealing to 580 °C.

## 4. Discussion

### 4.1. The Deformed State

The starting material was a hot-rolled thick plate of 2A14 alloy. Before the semisolid annealing experiment, the raw material went through the hot rolling process with a cumulative deformation of 90% at 400 °C. During the process, on one hand, the deformation energy was stored in the microstructure in the form of various crystal defects. On the other hand, with the input of heat, the softening process of dynamic recovery (DRV) and dynamic recrystallization (DRX) also occurred. Huang and Logé [19] introduced, in detail, the classification of various dynamic recrystallization (DRX) mechanisms based on phenomenology. Many factors affect the DRX mechanism, including the stacking fault energy (SFE), the TMP conditions, the distribution of the grain size, the chemical composition of the second phase, and the distribution and size of the second phase [26]. It is well known that Al is a face centered cubic (FCC) close packed structure with a high SFE [27,28]. When subjected to large plastic deformation processes, such as extrusion or rolling, metals with a high SFE tend to facilitate cross-slips to form substructures like cells or subgrains during the formation of dislocation cell subdivisions, rather than undergoing twinning and twin fragmentation [26,29]. The present findings seem to be consistent with other research which found that the deformation characteristics in hot-rolled 2A14 alloy were dislocation cross-slipping to form dislocation cells and subgrains that exist in the form of high-density LAGBs in the EBSD results (Figure 5). Cells or subgrains with a high density of LAGBs are the characteristic result of typical DRV and continuous dynamic recrystallization (CDRX). As stated by Sakai et al. [30], Al alloys with a high SFE are more prone to CDRX and DRV under the condition of large plastic deformation, resulting in the production of ultrafine grains. DRV and CDRX often occurred simultaneously, and high strain rate promoted the occurrence of CDRX. In addition, a few fine recrystallized (RXed) grains existed in the hot-deformation structure (Figure 3c), which is the characteristic result of typical discontinuous dynamic recrystallization (DDRX). As Huang and Logé [19] pointed out, coarse second particles can accelerate DDRX by particle stimulated nucleation (PSN), which is consistent with the results of this study, where the coarse second phase along the HAGBs provided the condition for DDRX nucleation. Additionally, Hu et al. [31] pointed out that dispersed fine particles tend to hinder boundary motion and slow down the CDRX process, also known as the Zener drag effect, which can be expressed as Equation (2):(2)P∝γGB×Vfr¯,
where *P* is the Zener pressure, $\gamma_{GB}$ is the grain boundary energy, $V_{f}$ is the volume fraction of the particle, and $\bar{r}$ is the mean size of the particles. The present findings seem to be consistent with the above description, whereby the dispersed, fine second phase hinders the migration of LAGBs (Figure 11a,b). Therefore, there was a mixed softening mechanism of DRV, CDRX, and DDRX during the hot rolling process of the 2A14 Al alloy. A high SFE promotes DRV and CDRX, while because of the PSN and Zener drag effect, the DDRX mechanism also exists in the hot-rolled 2A14 Al alloy.

### 4.2. The Second Phases in the 2A14 Al Alloy

The second phases in the alloy are mainly determined by the alloy composition, while the size and distribution state of the second phase are affected by both the TMP conditions and the heat treatment process. The 2A14 alloy can be strengthened by standard solid solution (SS) and artificial aging treatment. The strengthening phase is mainly the transition phases θ″ and θ′ of the equilibrium phase θ-Al_2_Cu. The aging precipitation sequence of the 2A14 Al alloy can be expressed as follows: Al (super saturated solid solution)→{atomic clusters (Al, Cu)}→{G-P zones}→{transition θ″ phase precipitates}→{transition θ′ phase precipitates}→{equilibrium θ-Al_2_Cu} [9,11,32,33,34]. The hot-rolled state used in this study can be regarded as the T4 heat treatment state. There was a certain solid solution after hot rolling and quenching, and then it was stored at room temperature for a long period of time to form the natural aging state. There were atomic clusters (Al, Cu) and G-P zones in the alloy, as shown in Figure 12b. The insoluble particles shown in Figure 12a,b are considered to be atomic clusters (Al, Cu) and G-P zones of Al-Cu according to the results reported by Han et al. [35]. The transition phases θ″ and θ′ were the smallest phases in hot-rolled 2A14 Al alloy that play pinning roles in dislocation, inhibiting the annihilation of dislocation and then inhibiting the nucleation of continuous recrystallization. In previous studies, the strengthening effect of the transition phase on the alloy caused by the inhibition of the dislocation slip is often emphasized, but the effect of the transition phase on recrystallization is rarely discussed. Jiao et al. [36] reported on the atomic-scale structure of the transition β″ and β′ phases. The atomic-scale structure of aging precipitates is small enough to hinder the slip of dislocation. Ding et al. [37] reported on the pinning effect of aging precipitated particles on the migration of LAGBs and found that aging precipitated particles inhibited the static recovery of the A5083 aluminum alloy.

In addition to the aging precipitates, there were greater precipitate contents of Fe, Mn, and Si, as shown in Figure 11a,b. Fe and Si elements are inevitably introduced into the metallurgy process of commercial alloys, in which it is easy to form hard and brittle intermetallic phases like Fe_3_Al and AlFeSi and weaken the properties of the alloy [38,39]. Generally, Mn will be added to aluminum alloy to form a soft, solid-solution MnAl_6_ phase to dissolve Fe and Si and form (Mn, Fe)Al_6_ or Al(MnFeSi) [40,41,42]. However, the 2A14 alloy is mainly composed of the Al matrix phase, and the relative volume fraction of other phases is too low to be detected by XRD (Figure 10). Therefore, in this study, only the Al matrix phase and the eutectic coarse phase θ-Al_2_Cu precipitated along the grain boundary were identified in the XRD results (Figure 10). The dispersed fine precipitates in Figure 11a,b are considered to be (Mn, Fe)Al_6_ or Al(MnFeSi) in accordance with the results provided by Engler et al. [40,41,42]. In addition, Qian et al. reported the effects of the Mn content on the recrystallization resistance of aluminum alloys [43]. The results showed that, with the addition of Mn, numerous dispersed Al(Mn, Fe)Si phase compounds were generated, and obvious resistance on recrystallization was exhibited compared with base alloy free of Mn and dispersoids. The second phase of this size will have an obvious Zener pinning effect on interfaces such as LAGBs and HAGBs, and the Zener pressure can be expressed by Equation (2). Kuang et al. [44] reported the Zener pinning pressure during the migration of the grain boundary that cast the Al-Mn-Fe-Si alloy. The alloy was resistant to recrystallization and was stable up to a temperature of 550 °C. Hu et al. [45] emphasized that solute drag and the Zener pinning effect retarded recrystallization and grain growth during the annealing process.

The largest second phase was the coarse particles of the eutectic equilibrium θ phase precipitated at HAGBs, as shown in the EDS results in Figure 11c,d and the XRD pattern in Figure 10. Wang et al. [46] reported the directional solidification of the Al-Cu alloy, where the solute distribution of the eutectic equilibrium θ phase was infected by the thermoelectric magnetic field and the eutectic θ phase was precipitated at the HAGBs. As reported by Huang and Logé [19], the coarse second phase often acts as the heterogeneous nucleation point of recrystallization and promotes the recrystallization process via the PSN mechanism. She et al. [47] investigated the relationship of PSN with the recrystallization of hot-extruded 7055 aluminum alloy, and found that coarse second phase particles were more likely to induce PSN recrystallization and produce PSN recrystallized grains with random orientations. In addition, coarse second phase particles had a pinning effect on the migration of HAGBs that inhibit grain growth during annealing.

### 4.3. The Recrystallization Mechanism

In order to more intuitively describe the recrystallization behavior of hot-rolled 2A14 aluminum during semisolid annealing, a schematic diagram of the recrystallization mechanism of the process was drawn, as shown in Figure 13. The whole process can be described in three stages: the hot-rolled microstructure state at room temperature, the recrystallization homogeneous nucleation state during continuous heating up to a near solidus temperature, and the static recrystallized (SRXed) state after heating up to a semisolid temperature. The hot-rolled state of the 2A14 Al alloy at room temperature (RT) had a complex microstructure. The grains were elongated along the rolling direction and in the coarse second-phase precipitates at HAGBs (equilibrium θ-Al_2_Cu phase). At the same time, there were dynamic recrystallized grains near the coarse second phase. Wang et al. [48] studied the DRX and phase distribution of the 7075 aluminum alloy during hot extrusion. The results show that the microstructure was richer in terms of the scale hierarchy under hot deformation when the CDRX mechanism was used, and the second phases were easier to refine and distribute in the microstructure under thermoplastic deformation. Behnamfard et al. [49] investigated the hot-rolling process of the aluminum matrix nanocomposite and found that the hot rolling process improved the mechanical properties of nanocomposites, probably because the ultrafine grain structure formed by hot rolling have an interactive strengthening effect with nanoceramics.

When the hot-rolled structure was annealed to a high solid-state temperature, firstly, the grain size increased under the migration of the large angle grain boundary, while the grain growth speed was slow due to the pinning effect of the coarse second phase on HAGBs. Homogeneous recrystallization occurred in some regions; that is, the annihilation of dislocation and the migration of the small angle grain boundary produced low-density grain boundary regions (see in Figure 5 and Figure 6a–c). The dislocation density was still very high in most areas, and the density of the LAGBs was still very high. This was because there were fine precipitates in the grains, the Zener drag effect hindered the migration of LAGBs, and there were atomic clusters (Al, Cu) and G-P zones that hindered the slip of dislocations. Hence, it could be conceivably hypothesized that the homogeneous nucleation zone was the precipitation free zones (PFZs), or there was a certain temperature fluctuation in the homogeneous nucleation zone, which made the second-phase particles and atomic clusters dissolve in the matrix. Chen et al. [50] investigated the precipitation behavior of the Al-Cu-Mg aluminum alloy and found that excessive Mg resulted in the formation of PFZs, which supports the hypothesis that the homogeneous nucleation zone occurs in the PFZs in 2A14 alloy. Wei et al. [51] also reported that irregular and uniform PFZs appeared in the Al-Mg-Si alloy after the aging process, which support this hypothesis. The report on PFZs also supports the results of Han et al. [35] regarding the aging precipitates of Al-Cu aluminum alloy.

When the annealing temperature reached the semisolid temperature, the dispersed fine precipitates were dissolved in the matrix, the insoluble atomic clusters were dissolved, the pinning effect on LAGBs and the blocking effect on the dislocation slip disappeared, the dislocations in the microstructure were annihilated, and the migration of LAGBs reduced the density of the LAGBs (see the decrease in the peak value of KAM in Figure 6b,c). The coarse second phase partially melted, the barrier effect on the large angle grain boundary migration decreased, and the grain grew rapidly.

## 5. Conclusions

In this paper, the SRX process of hot-rolled 2A14 Al alloy during continuous heating to the semisolid temperature was investigated. The XRD, EBSD, and TEM methods were applied to analyze the phase distribution, microstructure, and the SRX mechanism from various perspectives and various scale hierarchies. The following conclusions can be drawn:(1)The obvious recrystallization nucleation process only occurred when the temperature rose to the near solidus temperature, and the recrystallization process mainly occurred after the temperature rose to the semisolid temperature. The recrystallization mechanism was dominated by CSRX. With an increase in the annealing temperature, the grains grew slowly. After reaching the semisolid temperature, the grains grew rapidly.(2)Recrystallization did not occur until the annealing temperature was near the solidus because of the Zener drag effect of the dispersed precipitates on the LAGBs and the blocking effect of atomic clusters on the dislocation slip. The homogeneous recrystallization nucleates in the PFZs or the predissolution zone of the precipitates/atomic clusters.(3)With the process of semisolid annealing, the deformation texture fraction in the microtexture decreased, while the recrystallization texture fraction increased, and the microtexture changed from the preferred orientation to a random orientation.

## Figures and Tables

**Figure 1 materials-16-02796-f001:**
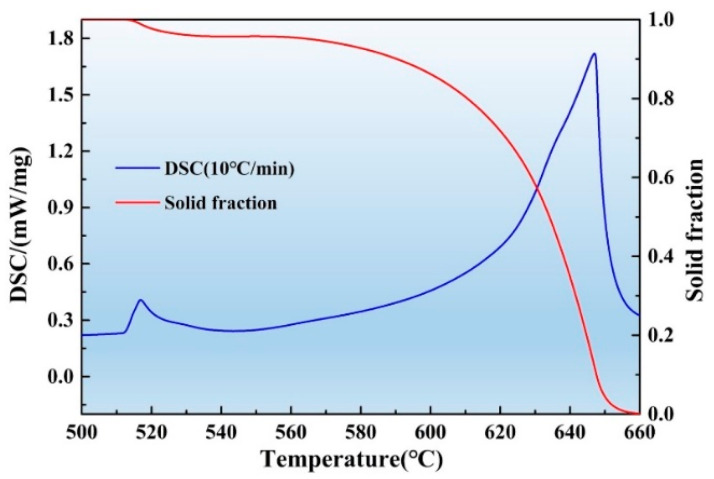
Semisolid range and solid fraction—temperature curve of the 2A14 Al alloy.

**Figure 2 materials-16-02796-f002:**
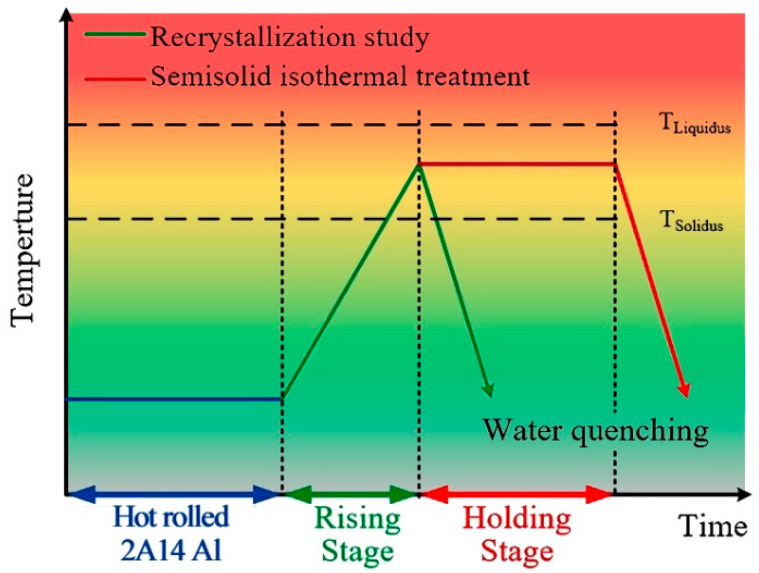
Schematic illustration of the WADSSIT process.

**Figure 3 materials-16-02796-f003:**
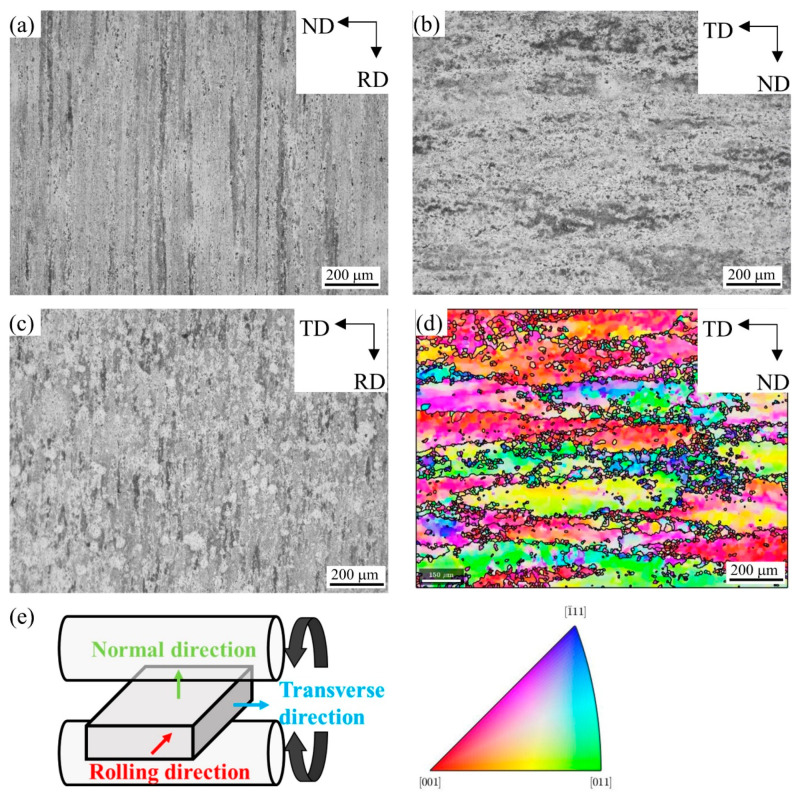
Microstructure of the raw hot-rolled 2A14 Al alloy on (**a**) the ND-RD plane, (**b**) the TD-ND plane, and (**c**) the TD-RD plane; (**d**) EBSD-IPF image of raw material on the TD-ND plane; (**e**) schematic diagram of the hot-rolled plate direction.

**Figure 4 materials-16-02796-f004:**
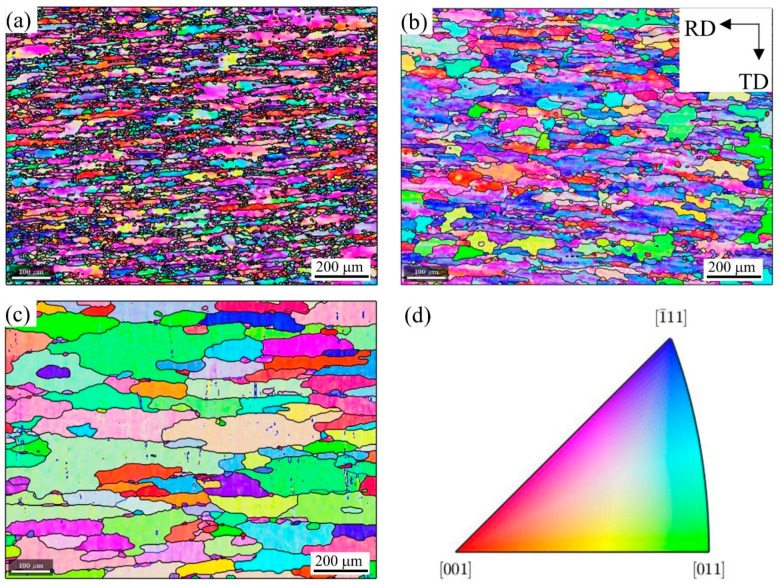
EBSD-IPF images of the 2A14 Al alloy on the RD-TD plane during annealing to (**a**) 505 °C, (**b**) 555 °C, and (**c**) 580 °C; (**d**) IPF grain orientation ruler.

**Figure 5 materials-16-02796-f005:**
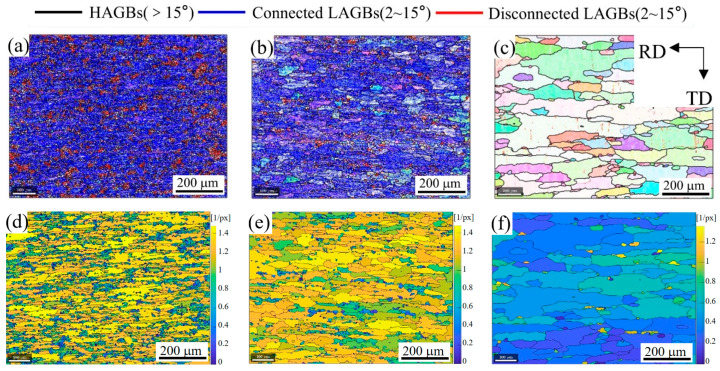
Grain boundary images of the 2A14 Al alloy on the RD-TD plane during annealing to (**a**) 505 °C, (**b**) 555 °C, and (**c**) 580 °C. Grain boundary density images of the 2A14 Al alloy on the RD-TD plane during annealing to (**d**) 505 °C, (**e**) 555 °C, and (**f**) 580 °C.

**Figure 6 materials-16-02796-f006:**
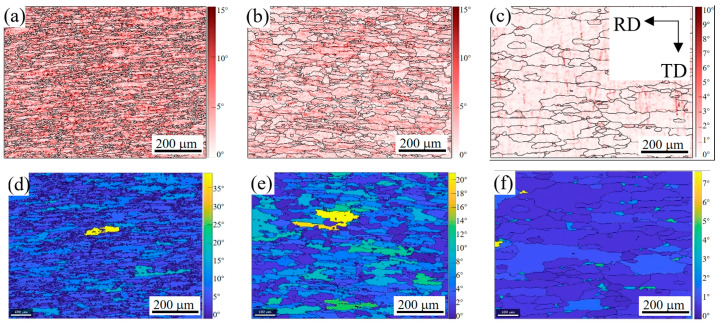
Kernel average misorientation results of the 2A14 Al alloy on the RD-TD plane after annealing to (**a**) 505 °C, (**b**) 555 °C, and (**c**) 580 °C. Grain orientation spread maps of the 2A14 Al alloy on the RD-TD plane after annealing to (**d**) 505 °C, (**e**) 555 °C, and (**f**) 580 °C.

**Figure 7 materials-16-02796-f007:**
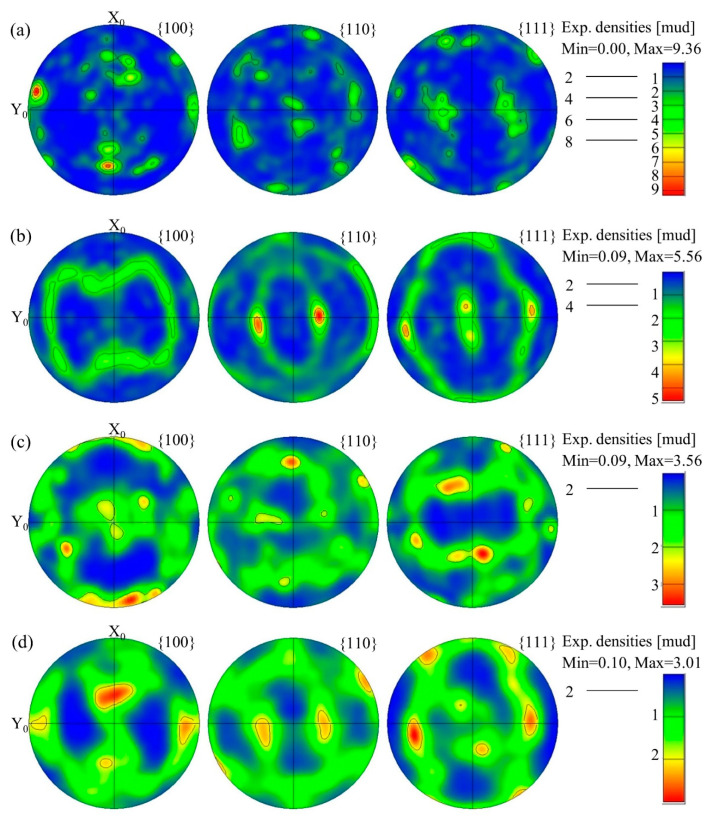
{100}, {110}, and {111} pole figures of the 2A14 Al alloy: (**a**) hot-rolled state after annealing to (**b**) 505 °C, (**c**) 555 °C, and (**d**) 580 °C.

**Figure 8 materials-16-02796-f008:**
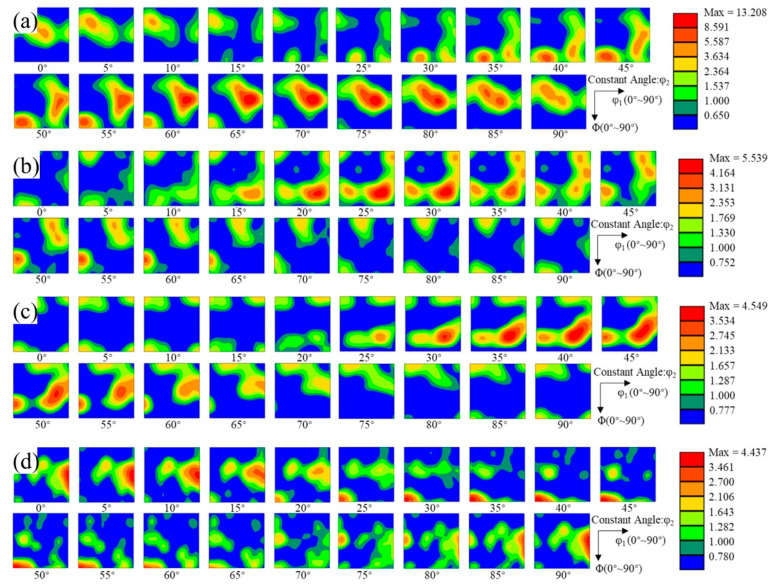
Orientation distribution function (ODF) maps where φ_2_ is a constant value and the interval of the 2A14 Al alloy is 5°: (**a**) hot-rolled, annealed to (**b**) 505 °C, (**c**) 555 °C, and (**d**) 580 °C.

**Figure 9 materials-16-02796-f009:**
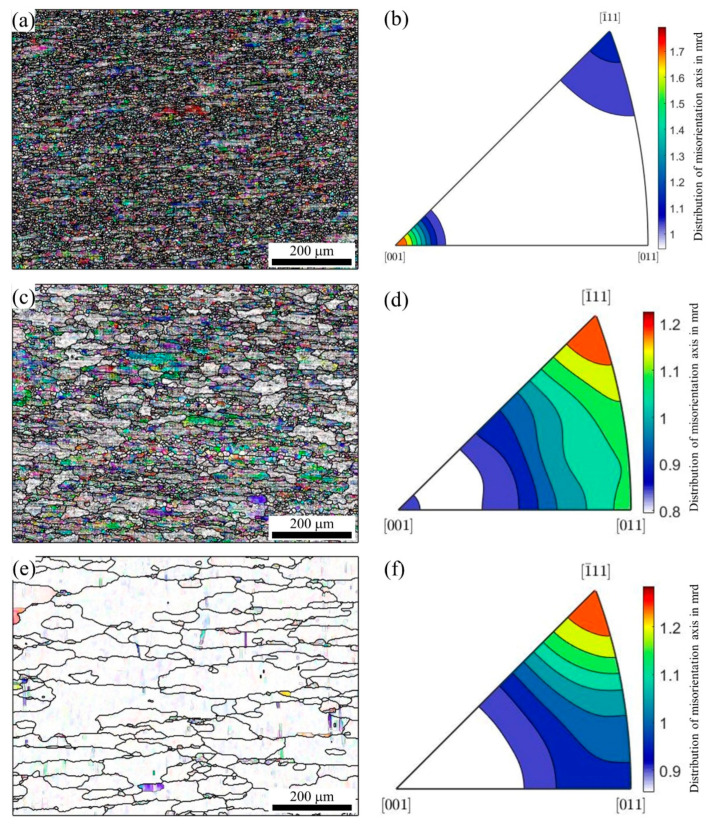
The misorientation axis in crystal coordinates and the corresponding distribution of the misorientation axis: (**a**,**b**) 505 °C, (**c**,**d**) 555 °C, and (**e**,**f**) 580 °C.

**Figure 10 materials-16-02796-f010:**
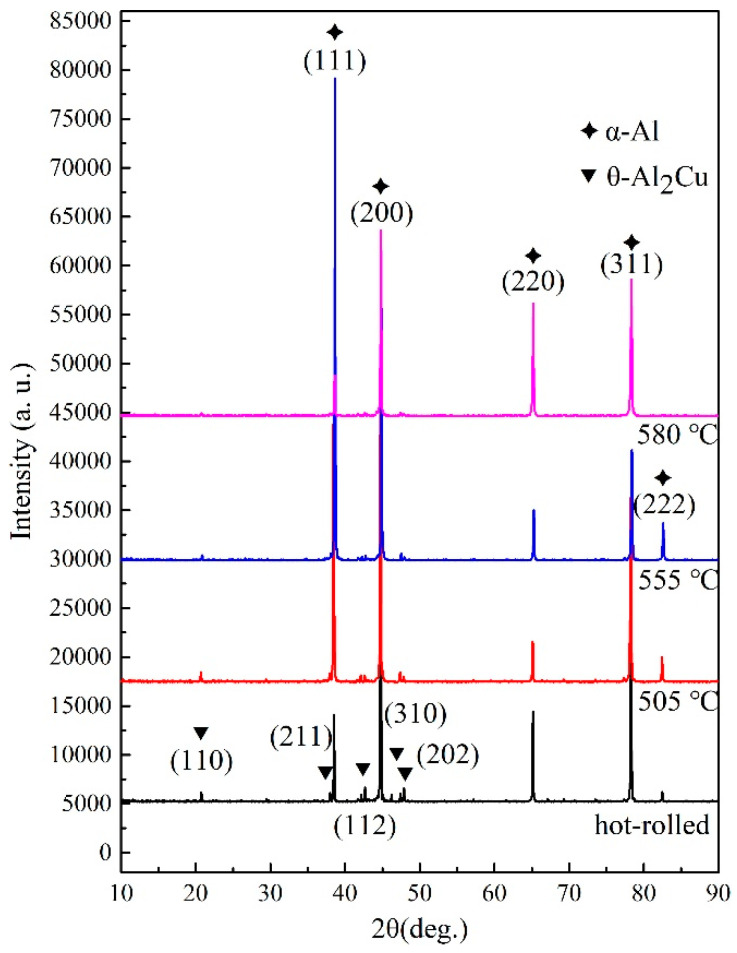
XRD result of the 2A14 aluminum alloy at the hot-rolled state after annealing to 505 °C, 555 °C, and 580 °C.

**Figure 11 materials-16-02796-f011:**
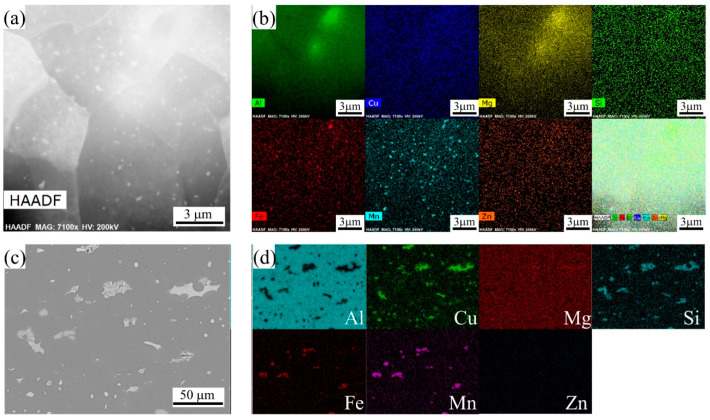
STEM images of 2A14 alloy after annealing to 555 °C: (**a**) HAADF image of the precipitates, (**b**) EDS mapping images of the main alloying elements; (**c**) SEM images of the 2A14 alloy, backscattered electron map and (**d**) the corresponding EDS mapping images of the main alloying elements.

**Figure 12 materials-16-02796-f012:**
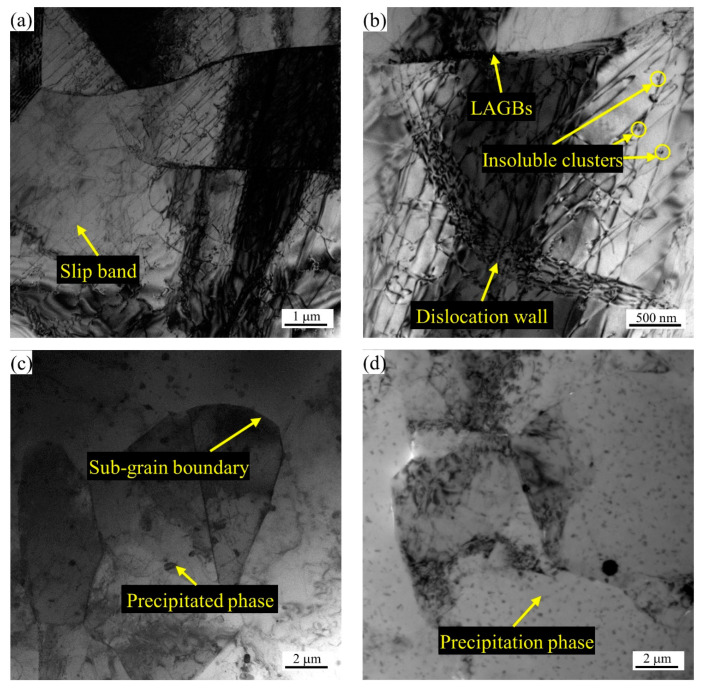
TEM bright field images of the 2A14 alloy after annealing to (**a**,**b**) 505 °C, (**c**) 555 °C, and (**d**) 580 °C.

**Figure 13 materials-16-02796-f013:**
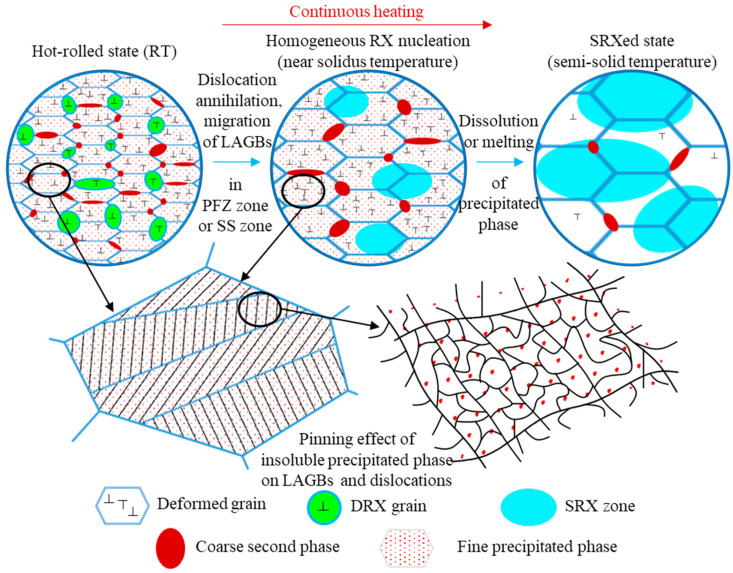
Schematic diagram of the recrystallization mechanism during semisolid annealing of hot-rolled 2A14 Al alloy.

**Table 1 materials-16-02796-t001:** Chemical composition of the 2A14 aluminum alloy (wt.%).

Element	Cu	Si	Mn	Mg	Fe	Ti	Zn	Ni	Al
Standard	3.9~4.8	0.6~1.2	0.4~1.0	0.4~0.8	≤0.7	≤0.15	≤0.3	≤0.1	Bal.
Sample	4.370	0.916	0.781	0.590	0.177	0.025	0.027	0.006	Bal.

**Table 2 materials-16-02796-t002:** Main textures of the hot-rolled 2A14 Al alloy.

Texture Type	Texture Name	{hkl}<uvw>	(φ_1_, Φ, φ_2_)
Deformation texture	Copper	{112}<111>	(90, 35, 45)
	Brass	{011}<211>	(35, 45, 0)
	S	{123}<634>	(59, 37, 63)
Recrystallization texture	P	{011}<122>	(70, 45, 0)
	Recrystallized brass	{236}<385>	(79, 31, 33)

## Data Availability

The data presented in this study are available on request from the corresponding author.

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
