# Peer review of "Recrystallization of Hot-Rolled 2A14 Alloy during Semisolid Temperature Annealing Process"

_materials, 2023, doi:10.3390/ma16072796_

Round 1

Reviewer 1 Report

The paper indicated that the increase of temperature, equilibrium θ-Al2Cu gradually dissolved in the matrix. It is a good work. The paper can be suggested after the revision.

-Please add a notation list.

- In order to provide a more comprehensive literature review, the authors should cite and discuss the following relevant paper in their revised manuscript:

Nassiraei, H., Mojtahedi, A., Lotfollahi-Yaghin, M.A. and Zhu, L., 2019. Capacity of tubular X-joints reinforced with collar plates under tensile brace loading at elevated temperatures. Thin-Walled Structures142, pp.426-443.

- In lines 45 and 46, we see “The supply status was hot rolled medium-thick plate of 50mm with a cumulative deformation of 90% at 400 °C”. Why was 400 °C selected?

-In line 488 “Fig. 11(a) and (b)” should be “Figs. 11(a) and (b)”. also line 509 for “Fig. 11 (c) and (d)”.

-Fig. 11 needs more discussion in the text.

- in line 241, the author said “the peak value of KAM decreased to 10, the KAM value of most areas decreased below 3”. It can be seen from fig. 6. The author should discus about the reason of this phenomenon. It is an important problem.

Author Response

Response to Reviewer 1 Comments

Point 1: Please add a notation list.

Response 1: According to the reviewer's suggestion, we have added a notation list in the appendix part.

Point 2: In order to provide a more comprehensive literature review, the authors should cite and discuss the following relevant paper in their revised manuscript:

Nassiraei, H., Mojtahedi, A., Lotfollahi-Yaghin, M.A. and Zhu, L., 2019. Capacity of tubular X-joints reinforced with collar plates under tensile brace loading at elevated temperatures. Thin-Walled Structures142, pp.426-443.

Response 2: As suggested by reviewer, we have added the citation in the 2nd paragraph of the introduction:"Nassiraei et al. [10] proposed a novel method to determine the ultimate capacity of collar plate reinforced weld X-joints at elevated temperatrues."

10.  Nassiraei, H.; Mojtahedi, A.; Lotfollahi-Yaghin, M.A.; Zhu, L. Capacity of tubular X-joints reinforced with collar plates under tensile brace loading at elevated temperatures. Thin-Walled Struct. 2019, 142, 426–443, doi:10.1016/j.tws.2019.04.042.

Point 3: In lines 45 and 46, we see “The supply status was hot rolled medium-thick plate of 50mm with a cumulative deformation of 90% at 400 °C”. Why was 400 °C selected?

Response 3: The rolling temperature was determined by the technical standards of the Northeast Light Alloy Co., Ltd of China. This was related to the recrystallization temperature of the material and the dynamic recrystallization softening of the rolling force during the process. The corresponding rolling temperature was optimized by the company's process, which was not what our research was concerned about. The description of rolling temperature in our article is to illustrate the hot working state of the starting material. The commercial supply state of wrought aluminum alloys is mostly hot rolled plates and hot extruded bars. This is applicable to our proposed semi solid short process billet making process.

Point 4: In line 488 “Fig. 11(a) and (b)” should be “Figs. 11(a) and (b)”. also line 509 for “Fig. 11 (c) and (d)”.

Response 4: We sincerely thank the reviewer for careful reading. As suggested by the reviewer, we have corrected the errors.

Point 4: Fig. 11 needs more discussion in the text.

Response 4: As suggested by the reviewer, we discuss more detail about Fig. 11 in section 4.1, and the 2nd and 3rd paragraphs in section 4.2:"In addition to the aging precipitates, there were a larger size of precipitates content of Fe, Mn and Si as shown in Figs. 11(a) and (b). Fe and Si elements are inevitably introduced into the metallurgy process of commercial alloys, which is easy to form hard and brittle intermetallic phases like Fe3Al, AlFeSi, etc." and "The largest second phase was the coarse particles of eutectic equilibrium θ phase precipitated at HAGBs, as shown in the EDS results in Figs. 11 (c) and (d) and the XRD pattern in Fig. 10."

Point 5: in line 241, the author said “the peak value of KAM decreased to 10, the KAM value of most areas decreased below 3”. It can be seen from fig. 6. The author should discus about the reason of this phenomenon. It is an important problem.

Response 5: According to the reviewer's suggestion, we have added the dicussion in the last paragraph of section 4.3: "When the annealing temperature reached the semi-solid temperature, the dispersed fine precipitates were dissolved in the matrix, the insoluble atomic clusters are dissolved, the pinning effect on LAGBs and the blocking effect on the dislocation slip disappeared, the dislocations in the microstructure were annihilated, and the migration of LAGBs reduced the density of the LAGBs (see the peak value of KAM decreased form Fig. 6(b) to (c)). The coarse second phase melted partially, the barrier effect on large angle grain boundary migration decreased, and the grain grew rapidly." Besides, in order to increase the correspondence between the discussion and the results in the manuscript, we have also added the following description in 2nd paragraph of section 4.2: "Homogeneous recrystallization occurred in some regions, that is, the annihilation of dislocation and the migration of small angle grain boundary produce low-density grain boundary regions (see in Fig. 5 and Fig. 6(a)-(c))."

Reviewer 2 Report

1. On the EBSD map in Fig. 3 there are numerous small grains. Authors are asked to comment on that - are these really such small grains, or is it a result of the cleaning procedure of the EBSD results? Because as can be seen, the map is completely cleaned and there are no unsolved points. What was the indexing rate of the EBSD measurements?

2. Fig. 4 - as in the previous point, what was the indexing rate of EBSD analysis?

3. Fig. 5 - there should be unit in the case of grain boundaries density.

4. Fig. 6 - lack of units on the maps.

Author Response

Response to Reviewer 2 Comments

Point 1: On the EBSD map in Fig. 3 there are numerous small grains. Authors are asked to comment on that - are these really such small grains, or is it a result of the cleaning procedure of the EBSD results? Because as can be seen, the map is completely cleaned and there are no unsolved points. What was the indexing rate of the EBSD measurements?

Response 1: Thank the reviewer for the professional question. The “small grains” are real grains and are sure not a result of the cleaning procedure of the EBSD results. Although we did conduct a noise reduction process on the EBSD map, no new grains with no-indexed were generated during the noise reduction process. Our noise reduction method was strictly follow the steps guided in MTEX (see website: https://mtex-toolbox.github.io/GrainReconstruction.html), that is, firstly (1) reconstruct the basic grains based on the grain determination threshold, and then (2) fill in the data of unresolved points. The reason why this process does not generate unresolved new grains is that during (1) grain reconstruction process, unresolved points are not filled, and the number of the grains has been fixed after completing the reconstruction. During the subsequent (2) filling process of unresolved points, the number of grains does not change, so this process will not generate unresolved new grains. The indexing rate of the EBSD in Fig. 3 was 83.96%. In fact, "small grains" were not too small. because the magnification in Fig. 3 was low, there were large grains of hundreds of microns in the hot rolled microstructure. Under the scale in Fig. 3, "small grains" were tens of microns big in size.

Point 2: Fig. 4 - as in the previous point, what was the indexing rate of EBSD analysis?

Response 2: The indexing rates of the EBSD results in Fig. 4 were 85.29%, 94.91%, and 93.92% at the annealing temperature of 505 ℃, 555 ℃, and 580 ℃, respectively. And we did conduct the same noise reduction process on the EBSD map as we answered in response 1 to improve the visualization and make sure the impact on subsequent analysis is minimized.

Point 3: Fig. 5 - there should be unit in the case of grain boundaries density.

Response 3: Thanks for the reviewer’s suggestion. We have added the unit in Fig. 5 as suggested by reviewer. As defined in MTEX (see the website: https://mtex-toolbox.github.io/SubGrainBoundaries.html), the grain boundary density is defined as sub-grain boundary length⁄grain area. Since the unit of length in the software is pixel, the unit in Fig. 5 should be 1⁄px.

Point 4: Fig. 6 - lack of units on the maps.

Response 4: According to reviewer's suggestion, we have added the units of "°" in Fig. 6. 

Round 2

Reviewer 1 Report

OK

Reviewer 2 Report

Authors answered to all my comments and I am satisfied with the answers.